# Synthesis of Silver Nanoparticles: From Conventional to 'Modern' Methods—A Review

**Ngoc Phuong Uyen Nguyen** [1], **Ngoc Tung Dang** [1], **Linh Doan** [2,3,*] and **Thi Thu Hoai Nguyen** [1,4,*]

1   School of Biotechnology, International University, Vietnam National University of Ho Chi Minh City,
    Ho Chi Minh City 700000, Vietnam; phuongnguyen20600@gmail.com (N.P.U.N.); dntung54@gmail.com (N.T.D.)
2   School of Chemical and Environmental Engineering, International University,
    Vietnam National University of Ho Chi Minh City, Ho Chi Minh City 700000, Vietnam
3   Nanomaterials Engineering Research & Development (NERD) Laboratory, International University,
    Vietnam National University of Ho Chi Minh City, Ho Chi Minh City 700000, Vietnam
4   Research Center for Infectious Diseases, International University,
    Vietnam National University of Ho Chi Minh City, Ho Chi Minh City 700000, Vietnam
*   Correspondence: dhlinh@hcmiu.edu.vn (L.D.); ntthoai@hcmiu.edu.vn (T.T.H.N.);
    Tel.: +84-28-37244270 (ext. 3335) (T.T.H.N)

**Abstract:** Silver nanoparticles, also known as AgNPs, have been extensively researched due to their one-of-a-kind characteristics, including their optical, antibacterial, and electrical capabilities. In the era of the antibiotics crisis, with an increase in antimicrobial resistance (AMR) and a decrease in newly developed drugs, AgNPs are potential candidates because of their substantial antimicrobial activity, limited resistance development, and extensive synergistic effect when combined with other drugs. The effect of AgNPs depends on the delivery system, compound combination, and their own properties, such as shape and size, which are heavily influenced by the synthesis process. Reduction using chemicals or light, irradiation using gamma ray, laser, electron beams or microwave and biological synthesis or a combination of these techniques are notable examples of AgNP synthesis methods. In this work, updated AgNP synthesis methods together with their strength and shortcomings are reviewed. Further, factors affecting the synthesis process are discussed. Finally, recent advances and challenges are considered.

**Keywords:** AgNPs; biological synthesis; chemical synthesis; physical synthesis; silver nanoparticles

## 1. Introduction

The increased interest in nanotechnology among researchers in recent years is not surprising because this field has advanced to unprecedented levels. Generally, nanotechnology is a multifaceted field that can be applied to electronics, sensors, optics, mechanics, catalysis, chemistry, cosmetics, pharmaceuticals, medicines and biomedical sciences, food technology and the environment [1–6]. Variation in chemical compositions, morphologies, or size, or controlled dispersities contribute to variation in characteristics of nanoparticles (NPs). This variation can be a result of the synthesis process, which is also affected by multiple factors. Today, production of NPs requires the produced particles to be nano-sized as well as their synthesis to be simple, inexpensive, environmentally favorable, and tailored to specific applications.

NPs, in general, have a size in the range of 1–100 nm [2]. Silver nanoparticles (AgNPs) stand out among most NPs because of their remarkable antimicrobial, electrical, optical, and thermal properties [1]. The antimicrobial mechanisms of AgNPs are complex, resulting in their broad-spectrum activity making it difficult for microorganisms to develop resistance, thus there are AgNP-resistant strains [7]. In addition, AgNPs' unique characteristic also enable wide application in various disciplines, including electronics, chemical/biological sensors, materials, and cosmetic and pharmaceutical products [1]. As such, the demand

for AgNPs has increased, requiring sufficient and reasonable supply. Therefore, synthesis of AgNPs has emerged as a critical issue in nanoscience. This work is to demonstrate synthesis methods of AgNPs, recent advances, their future challenges, and opportunities.

## 2. Synthesis of Silver Nanoparticles

Existing synthesis methods can be classified into two types using either a top-down or bottom-up approach (Figure 1).

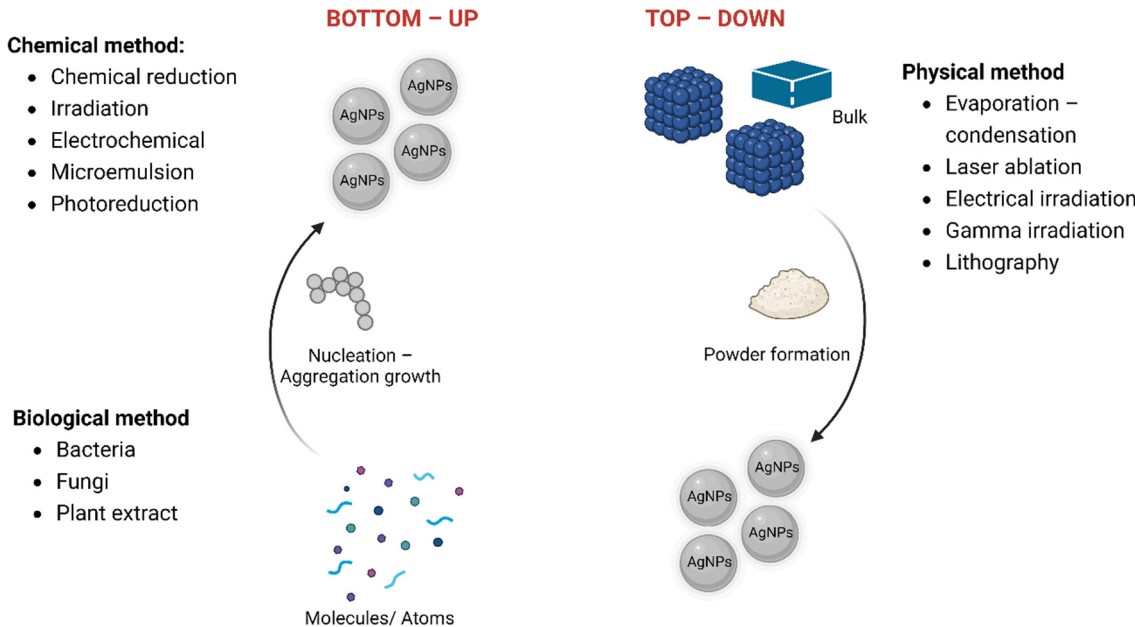

**Figure 1.** AgNP synthesis using a top-down and a bottom-up approach. In the top-down strategy, NPs are formed from bulk materials; in the bottom-up strategy, NPs are formed from molecular components. Created with BioRender.com.

In the top-down approach, metal NPs are formed from bulk materials through a variety of physical forces, including mechanical forces such as crushing, grinding, and milling; electrical forces such as electrical arc discharge or laser ablation; and thermal forces such as vapor-condensation. NPs produced using these methods can be devoid of chemical additives and have sizes ranging from 10 to 100 nm. Physically synthesized NPs are also more likely uniform in particle size distribution and possess a high purity. A weakness of this approach is that because it does not employ chemicals which are potentially detrimental to humans or the environment, it also does not have stabilizers or capping agents to prevent agglomeration [8–10]. In addition, these procedures require complex equipment and external energy [8].

In the bottom-up approach, complex aggregates are constructed from molecular components through nucleation and growth. Common bottom-up methods are chemical and biological synthesis to generate NPs from precursor salt. Chemical synthesis can be combined with lights, electricity, microwaves, and sonicwaves to enhance efficacy. Chemical synthesis is used to obtain NPs quickly in a variety of configurations. However, the use of potentially hazardous chemicals in the synthesis process hinders the application of these chemically generated NPs in medical applications. In recent decades, biological synthesis has become increasingly important due to its environmentally friendly property. Typically, biological synthesis uses molecules from microorganisms such as alcohols, flavonoids, alkaloids, quinines, terpenoids, and phenolic compounds, as well as exopolysaccharides, cellulose, and enzymes [8,11]. Biological synthesis is a cost-effective, dependable, environmentally favorable, and straightforward method. However, it also has weaknesses, which are discussed in the following sections together with details of methods.

### 2.1. Physical Methods

Laser ablation, irradiation, evaporation and condensation, and lithography are the most important physical processes for producing AgNPs [12]. Energy used in this approach can be light energy (laser ablation method), electrical energy (electrical arc-discharge method), thermal energy (physical vapor deposition) and mechanical energy (ball milling method). The benefits of these procedures are a fast synthesis reaction rate, high purity due to the lack of solvent contamination, and an absence of dangerous compounds (i.e., reducing agents/stabilizers). The physical synthesis of AgNPs employs an atmospheric pressure tube furnace to create NPs by evaporation–condensation [13–18]. However, this process has several disadvantages. For instance, a tube furnace requires a large amount of storage space, a substantial amount of energy for heating, and considerable time to achieve thermal stability [19,20]. Laser ablation of metallic bulk materials in solution, electrical irradiation of an Ag target in pure water with a 532 nm laser beam, or nanosphere lithography can create small NPs with a narrow size distribution in pure water without the use of any chemical additives, and the benefit of being a simple and low-cost nanofabrication approach [21–27]. Aside from the benefits highlighted above, some of the disadvantages of these techniques are poor potential yields and significant energy consumption [26,28–30]. Overall, physical methods generate AgNPs with a fine size distribution. They are suitable for the large-scale production of a singular product while producing AgNPs in ash form efficiently [31].

### 2.2. Chemical Methods

To date, the most common methods for producing AgNPs are still based on chemical synthesis. Under certain conditions, $Ag^+$ from a silver salt precursor, through electron transfer, is reduced to elemental silver (AgNPs) [32]. The processes of nucleation and growth take place to generate final the AgNP product. In brief, the concentration of silver element in the solution swiftly exceeds the supersaturation critical level, resulting in "burst nucleation" and precipitation, which leads to nucleus formation. Apart from nucleation, increased silver addition induces nuclei growth and the formation of larger NPs. These chemical AgNP synthesis techniques can produce NPs with no aggregation, a high yield, despite the high production costs and hazardous consequences [33,34]. In chemical synthesis procedures, the production of NPs requires three reactant components: a silver salt precursor, reducing agents, and a stabilizing chemical.

The function of stabilizing agents is to prevent agglomeration after synthesis. These reagents are usually surfactants containing functionalities serving as protectors, especially polymeric compounds, to coat AgNPs and protect the NP surface, prevent other AgNPs from absorbing on or binding to the NP surfaces leading to agglomeration [35]. However, Amir et al. demonstrated that the higher molar ratios for the stabilizer result in lower AgNPs due to an excess of stabilizer, preventing it from forming a complex with $Ag^+$ [36].

#### 2.2.1. Chemical Reduction

Numerous techniques have been devised for the chemical synthesis of AgNPs, including the chemical reduction method, the polyol method, and the radiolytic process. Chemical reduction using inorganic and organic reducing agents is the finest and simplest method for producing AgNPs without aggregation, with a high yield and minimal preparation cost [37]. Several reducing agents, including ascorbate, elemental hydrogen, polyol sodium borohydride (NaBH4), sodium citrate, and the Tollens reagent, are applied to reduce silver ions (Ag+) in aqueous or nonaqueous solutions. Principally, two elements are required for NP growth in this method: a silver salt and a reducing agent [38]. The silver ions are derived from silver salt such as silver nitrate, silver citrate and silver acetate. A reducing agent first reduces the ions to atoms: $Ag^+ (aq) + e^- \rightarrow Ag (s)$ [39], then atoms are nucleated and grow into particles. The concentration ratio of silver salt to reducing agent determines the availability of atoms, which governs the size and morphology of the NPs. Higher silver salt concentrations reportedly generate more NPs for 24 h reaction times but if the reaction

time increases to 96 h, AgNP population will not be homogenous. In this instance, the preponderance of Ag atoms is encapsulated within sizable NPs [40].

In case of mild reducing agent, the remaining $Ag^+$ ions continued generated to $Ag^0$ and firmly attached to the surface of existing Ag particles, causing the product's morphology to change, quasi-spherical to polygonal shape for instances [38]. In other words, a sluggish rate of reaction leads to particle agglomeration, whereas a stronger reducing agent produces smaller AgNPs [41].The appearance of black sediment can indicate that AgNPs undergo agglomeration [42]. Recently, scientists have evaluated the use of cationic exchange resins to separate free silver ions from suspensions of synthesized AgNPs, thereby reducing the Ag+ content of an unprocessed suspension of AgNPs while preserving their integrity [43].

### 2.2.2. Microemulsion Techniques

Microemulsion techniques refer to the synthesis of AgNPs using surfactant for dispersing two immiscible liquids, such as oil and water, water and superficial $CO_2$, a mixture of oil, one or few surfactants, and water [41]. This method can create homogeneous AgNPs with controllable size [35,44].

Basis for the preparation of AgNPs in two-phase aqueous organic systems is the initial spatial separation of reactants (Ag precursor and reducing agent) in two immiscible phases [45]. "Ready-to-use" surfactants could be anionic, cationic, zwitterionic, and non-ionic reagents including bis(2-ethyhexyl) sulfosuccinate, lauryl sodium sulphate, sodium dodecylbenzene sulfonate (SDS) (anionic), cetyltrimethylammonium bromide (CTAB) and polyvinylpyrrolidone (PVP) (cationic) and Triton X-100 (non-ionic) [46]. The selection of surfactants should be determined by the requirements of the experiment and the reaction conditions. Different surfactants, or microemulsion systems, used in the fabrication process will produce AgNPs with distinct diameters or morphologies. The rate of interactions between metal precursors and reducing agents is influenced by the interface between the two liquids as well as the intensity of interphase transport, which is mediated by ammonium salt. The formation of silver clusters at the interface is stabilized by the transfer of non-polar aqueous medium stabilizer molecules to the organic medium by the interphase transporter [35,41,47].

There are several influent parameters having ability to affect the shape or the size of AgNPs: the type of continuous phase, the amount of precursor dissolved within the nanodroplets, and the amount of water, referred to as the molar ratio of water to surfactant (W), etc. [48]. A large number of NPs with small diameters are created by the high exchange rate between micelles. In contrast, slow material exchange between micelles results in the formation of fewer nuclei and a larger ultimate particle size [49]. For instance, using borohydrate as a reducing agent and a biosurfactant extracted from *Pseudomonas aeruginosa* MKVIT3 resulted in the generation of cubic AgNPs having size of 17.89 and 8.74 nm [50]. Meanwhile, it was also reported that at 70 °C, a combination of silver acetate and the reducing agent oleylamine can yield highly monodisperse AgNPs, which have a size ranging from 10 to 20 nm and a storage stability of 6 months [41,50,51].

### 2.2.3. Photochemical Method

The photochemical process begins with metal precursors to be reduced from n+ valence ($M^{n+}$) to zero-valence ($M^0$) via the photocatalytic action of a reducing agent. The $M^0$ creates nucleation centers or nuclei, which proliferate and aggregate into metallic NPs [52,53]. Ultraviolet light, sunlight, and laser light are examples of light sources, with ultraviolet light being the most prevalent [8]. Photochemical synthesis permits the formation of NPs in a variety of media, including cells, emulsions, polymer films, surfactant micelles, and glassware [35]. Among the variables that can influence the synthesis of AgNPs are the light's source, intensity, and wavelength, as well as the irradiation duration. For instance, increasing duration and intensity of irradiation have been shown to promote Ag+ reduction [8,52,53].

Photochemical routes, as shown in Figure 2, in nanotechnology take more advantages over other methods, because they do not use toxic or hazardous compounds or require expensive equipment and highly trained personnel. They can be conducted at room temperature and atmospheric pressure [52].

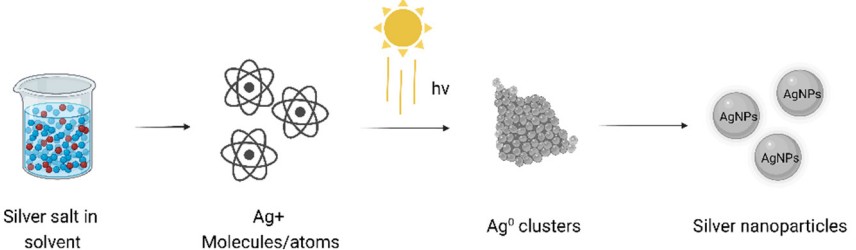

**Figure 2.** Photochemical formation of NPs [52]. Created with BioRender.com.

### 2.2.4. Polymers and Polysaccharides

To create AgNPs, water was used as an eco-friendly solvent, and polysaccharides were used as capping/reducing agents. Starch (the capping agent) and D-glucose (the reducing agent) were used to synthesize starch-AgNPs in a mildly heated system. Due to the feeble binding between starch and AgNPs, this linking is subsequently reversible at higher temperatures, and AgNPs are separated [35,53,54]. One of the most common polymers to use as reducing agents in synthesizing AgNPs is polyvinyl alcohol (PVA), polyethylene glycol (PEG), polyacrylamide, or other biopolymers such as chitosan [55–57]. PEG-coated AgNPs were reported to be extremely stable in extremely salty solutions, whereas carboxyl-coated lipoic acid particles can be used for bioconjugation [9]. In the case of PEG, ascorbic acid, thiosulfate, and sodium citrate were commonly used as reducing agents combination to create a specific shape of AgNPs, typically spherical AgNPs [41]. Meanwhile, stabilizing agents usually include polyvinyl alcohol (PVA), polyvinylpyrrolidone (PVP), bovine serum albumin (BSA), citrate, and cellulose [58].

Currently, several AgNP polymer hybridization strategies have been proposed to improve the antimicrobial activity of AgNPs. The most interesting and typical compound now is chitosan-AgNPs, since the chitosan's amine groups can alter bacterial cell permeability, improving the antibacterial activity of AgNP at low cost [59,60]. Because of metal ion coordination by the amino groups on the biopolymer chain, the principal amino groups of chitosan interact with metal surfaces and serve as capping sites for NP stabilization [59,61]. Silver ion reduction can occur via oxidation of other functional groups on the chitosan structure (e.g., hydroxyl groups), according to the mechanism of synthesis. The affinity between chitosan and AgNPs may be influenced by the formation of chemical bonds between nitrogen atoms, which are electron-rich elements, and the lone pairs in the silver orbitals [59,60]. Recently, a novel polyquaternary phosphonium oligochitosans (PQPOCs) linked with AgNPs. PQPOCs–AgNPs exhibited greater antiviral activity against norovirus (Nov), hepatitis A virus (HAV), and coxsackievirus B4 (Coxb4) than PQPOCs alone. The combination of PQPOCs-AgNPs is supposed to enhance molecular interactions with viral glycoproteins and block viral penetration [62,63].

### 2.2.5. Electrochemical Synthetic Method

In this method, AgNPs are formed under conditions of electrochemical discharge/ plasma in aqueous solutions in four main steps or reactions. Reaction 1, metallic ions are formed. In reaction 2, electrons are generated in microplasmas to readily hydrate in aqueous medium and can subsequently reduce metallic ions in reaction 3, so called electrochemical reduction. In reaction 4, nucleation of NPs takes place where metal atoms are combined into nanoclusters, then it is followed by growth and agglomeration that leads to the formation of NPs [64].

Reaction 1: $M^{n+} + ne^- \rightarrow M^0$
Reaction 2: $e^- + nH_2O \rightarrow e^-_{aq}$
Reaction 3: $M^{n+} + ne^-_{aq} \rightarrow M^0$
Reaction 4: $M^0 \rightarrow M_2 \rightarrow \ldots \rightarrow M_x \rightarrow \ldots \rightarrow M_{agglomerate}$

### 2.2.6. Microwave-Assisted Synthesis

Microwave-assisted synthesis was discovered for the first time in the early 1940s. The procedure entails swiftly heating the silver precursor with microwave irradiation, which may facilitate nuclei generation on-site. Thus, control of the nucleation and growth phases of AgNP synthesis is improved. Data showed that AgNPs produced by microwave-assisted synthesis have a narrow size distribution and a high degree of crystallization [41,65,66].

Several variables can impact the microwave-assisted synthesis of AgNPs. They include precursor concentration, stabilizer type and chirality of reducing agents. Water and alcohol are the optimal medium for microwave heating stabilizers due to their high dielectric losses. Polar molecules, such as water, attempt to align the electric field in a microwave. When dipolar molecules attempt to reorient themselves relative to an alternating electric field, they lose energy as heat, which may contribute to the reduction of $Ag^+$ [8,35]. Further, microwave power input, irradiation time, dielectric constant and medium refractive index can also influence the outcome of the synthesis process.

In addition to silver, microwave energy can also be used to synthesize silver-doped lanthanum chromites. It is possible to produce AgNPs by combining microwave energy and thermal reduction, which can then be deposited on oxidized carbon paper electrodes. This procedure yields AgNPs with homogenous particle sizes that are evenly distributed across a carbon paper substrate [8,35,67]. Despite of various chemical methods to synthesize AgNPs, each method has its own advantages and disadvantages, as shown in Table 1.

**Table 1.** Advantages and disadvantages of different chemical methods.

| No. | Method | Advantages | Disadvantages | References |
|---|---|---|---|---|
| 1 | Chemical reduction | Operate easily<br>Low cost | Toxic and hazardous chemicals | [68] |
| 2 | Microemulsion techniques | Low input of mechanical force<br>Theoretical consistency | Exceptionally susceptible to change<br>Extensive formulation effort<br>Low concentrations of AgNPs | [69] |
| 3 | Photochemical method | In situ highly fast dissolving AgNPs in the luminescence region<br>Utilize at ambient temperature<br>No dangerous or potent reducing agents<br>Not rely on costly equipment or highly trained personnel | Long time duration<br>expensive equipment<br>experimental environment | [52] |
| 4 | Electrochemical reduction | Metal ions come from sarcrificial anodes to reduce the quantity of precursors.<br>Simple reaction control, moderate reaction conditions, and less pollution | Unsuitable for large-scale AgNP production | [64] |
| 5 | Microwave-assisted method | Efficacy of energy conversion at a high level<br>Time-saving<br>Cleanliness, convenience<br>Produce on a large scale AgNPs with maximum dispersal | Expensive equipment<br>Unfeasible for reaction monitoring<br>Unsuitable for scale-up | [70] |

### 2.3. Green Chemistry Approach for the Synthesis of AgNPs

Chemical remnants of the solvent are frequently found on the surface of the synthesized AgNPs such as ethylene glycol, sodium citrate, oleyl amine, liquid paraffin. As

hazardous chemical carriers, these NPs are especially hazardous when used for drug delivery, antimicrobial action, or any other application requiring insertion of the NPs inside the human body [71]. The chemical synthesis of AgNPs on a large scale is not feasible in a world seeking to attain sustainable development objectives. Due to the danger of chemical methods, it is urgent to develop an alternative synthesis pathway that is economical and eco-friendly, biological or "green" method, as shown in Figure 3. Green synthesis can be classified as (a) the use of microorganisms such as fungi, yeasts (eukaryotes), bacteria, and actinomycetes (prokaryotes), (b) the use of vegetation and plant compounds, and (c) the utilisation of templates such as membranes, virus DNA, and diatoms [72].

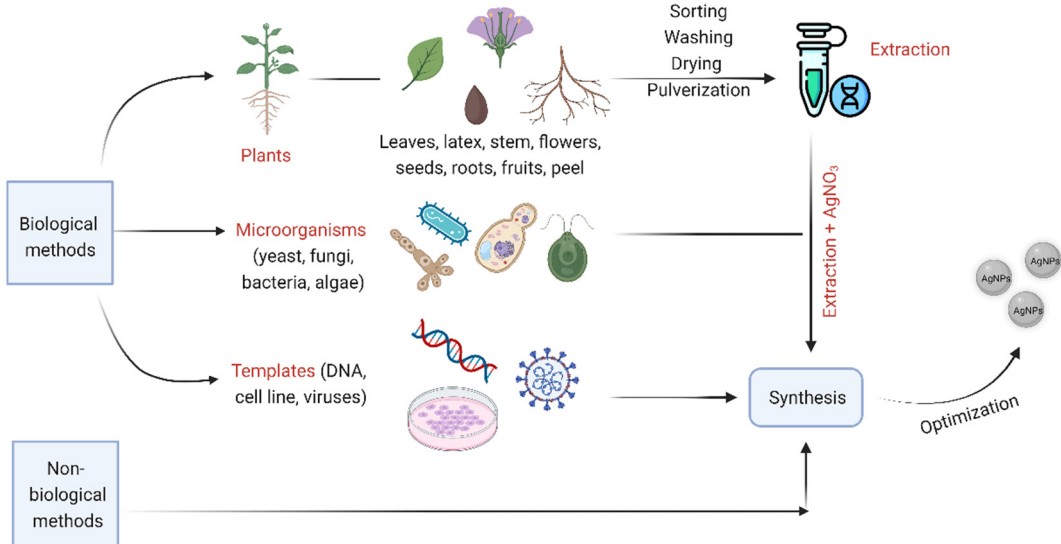

**Figure 3.** Biological approaches in AgNP synthesis. Created with BioRender.com.

For the ecological synthesis of AgNPs, a reducing biological agent is applied. In the majority of instances, different cell constituents serve as stabilizing and capping agents, eliminating the need for external capping and stabilizing agents [73,74].

The green synthesis mechanism can be described briefly as follows: Organic chemicals donate electrons for the reduction of $Ag^+$ ions to $Ag^0$. These organic chemicals can be proteins, enzymes & coenzymes, lipids, carbohydrates, alkaloids, flavonoids, phenols, terpenoids, and others. The active ingredient responsible for $Ag^+$ ion reduction varies depending on the used organism or biological extract. For the nano-transformation of AgNPs, electrons are believed to be derived from dehydrogenation of acids and alcohols in hydrophytes, keto to enol conversions in mesophytes, or both mechanisms in xerophytes. Similar reduction processes can be performed by microbial cellular and extracellular oxidoreductase enzymes [74].

Mohan and colleagues described a simple method for synthesis of AgNPs using Acacia gum (0.5% $w/v$) in a mild environment [75]. In this method, the carboxylate group of the acacia gum glycoprotein was converted into COOAg through an ion-exchange process then acacia gum polysaccharide polymer chains will facilitate the stabilization of formed AgNPs [76].

### 2.3.1. Plants

Plant-based synthesis of AgNPs is widely used in comparison to microorganism-based techniques because it is more effective, less bio-compromising, and does not require active cell cultures. Various plant parts such as bark, callus, flower, fruit, leaves, peel, rhizome, stem, and seed can be extracted to synthesize AgNPs [8,9] and AgNPs' shapes produced from plants are commonly spherical or oval [77]. They contain a high concentration of carbohydrates, enzymes, flavonoids, polyphenols, and proteins. In the cell- free metallic NPs synthesis, these phytochemicals are extracted and used directly as reducing and

stabilizing agents, thus supplanting potentially hazardous compounds such as sodium borohydride (NaBH4). Due to the presence of an extensive variety of phytoconstituents in the extracts, the precise mechanism underlying this phenomenon remains unknown. Although organic acids, polyphenols and proteins are believed to be the primary reducing agents, it is anticipated that the numerous phytochemicals collaborate. In general, this method is considered as a cost-effective solution for mass production [78–80]. Some of the active phytochemicals that may be responsible for the reduction of $Ag^+$ to AgNPs are terpenoids, polysaccharides, phenolics, alkaloids, flavones, amino acids, alcoholic compounds, enzymes, and. With non-pathogenic nature, and eco-friendly reaction conditions, highly economical single step protocol, synthesizing AgNPs using plant extracts in green synthesis have been identified to be speedier than that of microorganisms such as bacteria and fungi and as an ideal candidate [72,77]. Furthermore, AgNPs synthesized from plants seemed to have higher bioactivity than the chemically synthesized one. Sreelekha et al. (2021) synthesized AgNPs from *M. frondosa* leaf extract and sodium citrate, DPPH (1, 1-Diphenyl-2-picrylhydrazyl) assay was used to evaluate the antioxidant activity of two methods. Green NPs exhibited 91% scavenging activity at a concentration of 5 μg/mL, whereas chemically synthesized NPs exhibited only 79% activity at the same concentration. The antioxidant properties of flavonoids and phenolic compounds—present on the surface of green synthesized NPs—make them useful for the prevention and treatment of degenerative diseases [81]. Moreover, AgNPs from *Brachychiton populneus* showed a significant effect on cancer cell, specifically, on the U87 and HEK 293 cell lines. As the concentration increases, the proportion of viable cancer cells decreases [82].

### 2.3.2. Microorganisms

Bacteria

Bacteria are of great interest for NP synthesis, even though this process is fraught with difficulties, such as culture contamination, long procedures, and limited control over NP size. Due to their exceptional ability to reduce heavy metal ions, microbes are regarded as one of the most promising candidates for NP synthesis [77,83]. Bacterial ability to survive in an extremely silver-rich environment may contribute to AgNP accumulation [84,85]. Because of the ease of recovery of AgNPs, the extracellular method is preferable to the intracellular method [8]. Exopolysaccharide, peptides, reductase, cofactors, c-type cytochromes, and silver-resistant genes are examples of organic substances that can be used as reducing agents in bacteria. Bacteria which have been used to synthesize AgNPs includes *Lactobacillus bulgaricus* [86], *Rhodococcus*, *Brevundimonas* and *Bacillus* [87], *Staphylococcus aureus* [88], and *Escherichia coli* [89]. Several enzymes, including nitrate reductase and lactate dehydrogenase, have been implicated in the formation of AgNPs. Special amino acids such as arginine, cysteine, lysine, methionine may adhere to the surface of the particle and function as reducing agents [8].

Algae and Fungi

Algae grow quickly, are simple to harvest and can be scaled up easily. Algae are rich in pigments, peptides, proteins, and secondary metabolites [8,41]. With abundance of organic compounds, algae are an excellent candidate for AgNP biosynthesis [90]. These active organic matters can be used as reducing agents to create spheres [91], triangles [92], cubes [93], hexagons [94] AgNPs, etc.

The ability of fungi to decompose organic matter is well known using a variety of enzymes and molecules [95]. A large number of fungal proteins and enzymes, including protease, cellulase, chitinase, and glucosidase, can directly accelerate and increase AgNP synthesis. NADH and NADH-dependent nitrate reductase are involved in the reduction of $Ag^+$ ions to $Ag^0$ by fungi. Fungi can also synthesize AgNPs intra- or extracellularly with no toxic chemicals required. NPs produced by intracellular synthesis are smaller than those produced by extracellular synthesis. Intracellular synthesis is preferable for several reasons, including fewer purification steps needed. Filamentous fungi are of higher interest



than other fungi because AgNPs they produce have favorable morphological properties, stability, and a broad range of applications [41]. Depending on the fungus used, parameters such as agitation, temperature, light, and culture and synthesis periods can be altered to achieve the desired NP characteristics [96].

## 3. Factors Affecting Silver Nanoparticle Synthesis and Their Stability

### 3.1. Temperature

When the synthesis reaction is performed at high temperatures, the reaction rate significantly increases, which leads to the formation of larger particles and the onset of an undesirable reaction. At 50 °C, this study showed that the reaction mixture rapidly turned brown, and the end-product was a black powder. TEM of the formed AgNPs revealed large silver particles that are not spherical and appear to be linked [1]. In addition, when the temperature increased from 90 °C to 110–120 °C, the maximum surface plasmon absorption band was sharpened indicating the higher concentration of synthesized AgNPs [97]. However, when the reaction was conducted at 10 °C, the rate of reaction was very sluggish, and the color of the solution did not change until 3 h later [1,97]. Although many studies have reported quicker rates of synthesis at higher temperatures, it is essential to consider NP quality. Guilger-Casagran et al. reported that some fungal species' ability to produce NPs at high temperatures is because electrons can be transferred from liberated amino acids to silver ions. Nevertheless, extremely high temperatures between 80 and 100 °C could denature the proteins that make up the NP encapsulation. This denaturation modifies the nucleation of Ag+ ions, resulting in the aggregation and enlargement of NPs. Unsuitable conditions result in an increase in NP size and a loss of stability due to the reduced enzyme activity involved in the synthesis [96].

### 3.2. pH

Ondari Nyakundi et al. reported that AgNP synthesis was optimal at pH 7 [98]. The amount of synthesized AgNPs increased with the increase in the pH of the reaction with an ideal pH of 7.6. Because Ag is an anion in aqueous solutions, the interaction between Ag and biomass is ionic. At low pH levels, the biomass may contain more positively charged functional groups, permitting the Ag ions to approach the binding sites. Silver, a soft metal, attaches to biomass predominantly through amino and sulfhydryl groups, which are considered as soft ligands and bear a more positive charge at low pH values, making them able to bind and reduce $Ag^+$ to $Ag^0$. Moreover, carboxylic groups, which are prevalent in biomass, are protonated at a low pH and may contribute to the binding of Ag ions, even though this group is regarded as a difficult ligand [98].

### 3.3. Time

The size of AgNPs increased as the incubation period lengthened [98,99]. Scientists discovered that the wavelength of the peak (max) has red shifted by increasing the incubation period, which is mostly likely due to an increase in colloidal AgNPs resulting in increased absorbance.

### 3.4. Pressure

Pressure plays an important role in most NP physical synthesis methods. When producing NPs, increasing the pressure causes the particles to grow in size. NPs go from being round to cube shaped to cauliflower shaped as the pressure is increased [100]. On the other hand, Zikmund et al. (2023) reported on the high-pressure aggregation chamber in which AgNPs are formed. However, their data showed no clear relationship between NP size and the gas pressure in the reaction chamber utilized in that investigation. The NPs spent less time in the aggregation chamber due to the increased pressure, which also resulted in increased Ar (argon) flow. Therefore, the effect of greater pressure can be mitigated by decreasing the necessary duration [101].

### 3.5. AgNO₃ Concentration

The precursor, silver nitrate $AgNO_3$, serves a vital purpose determining the size of AgNPs. There are several studies that put the concentration of $AgNO_3$ into investigation to study the influence that this factor has, particularly on the size of AgNPs that are formed. In Htwe et al. (2019)'s study, by increasing the silver nitrate concentration from 0.5 mM to 0.7 mM and then to 0.9 mM, the average crystallite size of AgNPs grew from 23.87 nm to 24.51 nm and then to 25.16 nm. According to their findings, the average size of the NPs formed is proportional to the original concentration of metal ions, therefore increasing the initial concentration of metal ions resulted in larger NPs [102]. These conclusion is consistent with other authors [103,104].

### 3.6. Other Factors

The medium in which AgNPs are immersed has a significant impact on their long-term stability. The medium affects chemical reactivity and interparticle forces, which in turn affects aggregation, morphology, size and long-term stability ('aging') of the NPs, especially if AgNPs are stored or used in aqueous conditions. Controlling the stability of inorganic NPs is dependent upon their surface modification and functionalization [105].

According to aforementioned, stabilizers (or capping agents) help prevent other AgNPs that can be absorbed on or bind to NP surfaces leading to agglomeration. Scientists have functionalized AgNPs with other compounds to improve their stability such as polyethylene glycol (PEG), chitosan, PVP, PVA, polystyrene, gold (core-shell), chromium (alloy) or other polymers [105,106]. Figure from Lufsyi Mahmudin et al. [97] despicts that the addition of the stabilizer sharpened the absorption spectrum curves and resulted in a red shift in the maximum absorption peak at 429.01 nm for PVA and 427.09 nm for PVP; while the absorption peak of AgNPs without a stabilizer is not high and the spectrum is very wide. Additionally, scientists compared the stability of AgNPs in different solvents including cell culture media (Bold's Basal Medium—BBM), NaCl, acetone, and reagent alcohol. It was discovered that AgNPs were stable in cell culture media for more than 300 days with no decrease in the SPR band intensity as well as in other solvents [107,108]. Hence, it can be inferred that for AgNP long-term storage, the biopolymers present in the cell culture media may play an important role [107,108].

NP shells are chemically and physically robust structures that uniformly cover the NP core, in addition to linking AgNPs with stabilizers. Despite the absence of covalent bonds, shells are not as readily removed or replaced as ligands. In addition, shells provide an additional chemical environment not only for stabilizing the core but also for incorporating the desired functionality into the NPs. Typically, the shell is formed by chemical interactions or bonding between the shell's chemical constituents. In many instances, ligand-stabilized NPs are synthesized first, followed by the formation of shell structures by ligand exchange [86]. Different AgNPs with coating shells, such as metal-AgNPs, pistachio-AgNPs, silica-AgNPs, have been evaluated by scientists [109–111].

Zeta potential is a common unit to indicate the stability of nanomaterials. Due to charge equilibrium, suspensions with zeta potentials less than 20 mV are considered unstable and have a low NP aggregation capacity [112]. Within the median values of 40–60 mV, the zeta potential exhibits the stability factor and is considered viable in both positive and negative modes of measurement. It is believed that smaller, spherical particles and nano-preparations are more stable, and the zeta potential, a dispersion stability indicator, and the other parameters of pH, ionic strength, medium concentration, and polydispersity all contribute to the stability of nano-systems composed of metallic NPs [113].

## 4. Recent Advanced Synthesis Methods and Future Challenges

### 4.1. Recent Advanced Synthesis Methods

Due to the incredible properties of AgNPs, there is an increasing application in various fields, making AgNPs demands become worldwide and in large quantities. To fulfill this need, a lot of improvements have been studied to produce a sustainable source

of AgNPs that is non-toxic, eco-friendly, cost-effective, high quality and high quantity. Wojnicki et al. [114] used the microdroplets as a reactor to decrease AgNPs in size and size distribution in comparison with a typical batch reactor from $4.8 \pm 1.3$ nm to $2.5 \pm 0.5$ nm. The particles are smaller than the diameter of the segmented flow channel and, as a result, do not adhere to the channel surface via a very thin liquid capping layer [114]. Aiming to produce AgNPs in large-scale by bacteria with uniform size, scientists have transformed metallothionein gene of *Candida albicans* into *Escherichia coli* DH5$\alpha$ cells to improve *E. coli* tolerance to $Ag^+$ and increased yield of AgNP synthesis [89]. More interestingly, scientists have started to study NPs' synthesis from viruses. The viruses are made up of nucleoprotein particles. The proteins that make up a virus's capsid act as a highly reactive surface for metal ions to interact with [115,116]. Recently, there has been a surge in interest in using viruses to perform biosynthesis of NPs though AgNPs, on the other hand, are still in the development stage. Yang et al. [117] reported the successful biosynthesis AgNPs (2–9 nm) from tobacco mosaic virus (TMV). The amino acids in TMV contain various functional groups such as thiol, carboxyl, and hydroxyl, which typically act as reducing agents for Ag ion reduction [117,118].

*4.2. Future Challenges*

Various techniques, including chemical, physical, and photochemical processes, are utilized to synthesize distinct AgNPs. The green synthesis of AgNPs is the most desirable due to their simplicity, low cost, ease of operation, and efficiency. Since bacteria, fungi, and other creatures are commonly used in production, the synthesis process requires isolation of productive strains, which necessitates some complex steps. This type of procedure is frequently difficult in terms of retaining the standard culture in comparison to the chemical and physical conditions. Still, the identification of the best approach for increasing product yield efficiency so that the biological approach can be used on a larger scale is necessary, imminent, and important. The properties of AgNPs are remarkably dependent on their morphology [31].

Recently, the impact of AgNPs on the environment and human health has prompted scientists to focalize on the toxicity generated by these particles. When AgNPs are oxidized, silver ions are released, which are harmful to both human health and the environment. Furthermore, the ionic strength of the environment might alter the stability of AgNPs, promoting agglomeration, particularly in acidic pH. Because of all the changes that have occurred in the environment, as well as variations in toxicity and bioavailability, determining the hazards presented by AgNPs to human health and the ecosystem is vital [119].

Overall, the environmental impacts of AgNPs are unknown [120]. However, the silver ion ($Ag^+$(aq)) in water is highly toxic to prokaryotes, freshwater and marine invertebrates and fish due to the interaction between silver ions with various ligands, other metals, and bioaccumulation [120–125].

Moreover, AgNPs are also dangerous to mammalian cells originating from the skin, liver, lung, brain, vascular system, and reproductive organs—according to the outcomes of in vitro investigations through different routes of exposure [126]. The bioavailabilty and toxicity of AgNPs is closely linked to their form, size, concentration, aggregation, chemical coating, surface charge, phase transitions and the techniques utilized to synthesize them (biological, physical, or chemical routes). The toxicity of AgNPs is also affected by the targeted living organisms, which is related to the organism's defense systems for eliminating unwanted substances [127,128].

However, natural organic matter (NOM) can be adsorbed on AgNPs, thereby reducing agglomeration. NOM adsorption on AgNPs reduces $Ag^+$ release in the environment by blocking oxidation sites, and NOM acts as a reducing agent in the reversible formation of $Ag^+$ from $Ag^0$ owing to humic and fulvic acids. Reduce toxicity by decreasing the concentration of $Ag^+$ ions [129,130]. In addition, in aquatic and soil ecosystems, AgNPs can easily react with sulfide (sulfidation process) to generate $Ag^0/Ag_2S$ core-shell particles,

lowering their toxicity [128]. By regulating their intrinsic characteristics via synthesis and the environment parameters, the toxicity of AgNPs could be controlled.

Despite the uncertainty of adverse effects on human health, ecosystem, and environment, application of AgNPs for medical usage was researched intensively recently. In addition to antimicrobial activity, AgNPs have broad anticancer action that varies depeding on size, dose, concentration, and treatment duration. The smaller AgNPs can cause higher endocytosis thus higher cytotoxicity and genotoxicity. Spherical AgNPs have higher cytotoxicity than other shapes due to their larger surface-to-volume ratio. A larger AgNPs dose usually causes more apoptosis than a smaller one [8].

## 5. Conclusions and Outlooks

AgNPs can serve as a potential antimicrobial agent due to their unique characteristics, especially their properties of antimicrobial activity, antifungal activity, limited resistance development, extensive synergistic effect when being combined with other drugs and, importantly, low toxic effects on healthy human cell lines, low allergic reaction, and good tolerance. AgNPs are generated via physical, chemical, and biological synthesis processes. Over time with improved methods, AgNPs are continuously produced with higher productivity and stability. Because each synthesis method has certain advantages and disadvantages in size control, antimicrobial activity, toxicity, environmental concerns of each individual's necessity, selectivity of synthesis methods should be considered. Other parameters such as temperature, dispersion agent, and surfactant had a significant impact on the quality and quantity of the produced AgNPs. Further, the modification and functionalization of AgNPs also contribute to their activity. To sum up, despite intensive and extensive research on AgNPs, further research should be continued, especially on the biosynthesis and the clinical toxicity of AgNPs.

**Author Contributions:** N.P.U.N., N.T.D., L.D. and T.T.H.N. conducted the extensive literature review and wrote this article. All authors have read and agreed to the published version of the manuscript.

**Funding:** This research received no external funding.

**Institutional Review Board Statement:** Not applicable.

**Informed Consent Statement:** Not applicable.

**Data Availability Statement:** All data that support the findings of this study are included within the article.

**Conflicts of Interest:** The authors declare no conflict of interest.

## Abbreviations

| | |
|---|---|
| NPs: | nanoparticles |
| AgNPs: | silver nanoparticles |
| AMR: | antimicrobial resistance |
| SERS: | surface-enhanced Roman spectroscopy |
| PVA: | polyvinyl alcohol |
| BSA: | bolvine |
| PVP: | polyvinylpyrrolidone |
| PEG: | polyethylene glycol |
| SDS: | sodium dodecyl sulfate |
| TMC: | tobacco mosaic virus |

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
