# Peer review of "Synthesis of Silver Nanoparticles: From Conventional to ‘Modern’ Methods—A Review"

_processes, doi:10.3390/pr11092617_

Round 1

Reviewer 1 Report

In this Manuscript, the authors reported the “Synthesis of silver nanoparticles: from conventional to ‘modern’ methods.

1) The review is well written, and easy to follow.

2) This review focused on development of recent synthetic methods of silver nanoparticles while discussing the advantages and disadvantages of each method.

3) I found no issues in this review 

I believe this review is suitable for publication in Processes

Author Response

Reviewer 1:

In this Manuscript, the authors reported the “Synthesis of silver nanoparticles: from conventional to ‘modern’ methods.

1) The review is well written, and easy to follow.

2) This review focused on development of recent synthetic methods of silver nanoparticles while discussing the advantages and disadvantages of each method.

3) I found no issues in this review 

I believe this review is suitable for publication in Processes

Responses: Thank you very much for your positive comments!

Reviewer 2 Report

The present manuscript describes physical, chemical, and biological techniques for synthesizing AgNPs  together with their advantages and disadvantages, factors affecting the synthesis as well the recent developments and prospective advantages of AgNPs synthesis industry.

The manuscript “Synthesis of silver nanoparticles: from conventional to ‘modern’
methods – A Review" is very well written and easy to understand.

Only minor errors should be corrected such as:

Page 4:  

-          Alcohol in sentence “Typically, the biological method relies on macromolecular substances found in bacteria, fungi, and algae, such as enzymes, alcohol, flavonoids…” should be changed to plural

Page 8: Polymers and polysaccharides

- Delete the hyphen before D-glucose in the sentence “Starch (the capping agent) and -D-glucose (the reducing agent)….”

With kind regards

Author Response

Reviewer 2:

The present manuscript describes physical, chemical, and biological techniques for synthesizing AgNPs  together with their advantages and disadvantages, factors affecting the synthesis as well the recent developments and prospective advantages of AgNPs synthesis industry.

The manuscript “Synthesis of silver nanoparticles: from conventional to ‘modern’
methods – A Review" is very well written and easy to understand.

Only minor errors should be corrected such as:

Page 4:  

-          Alcohol in sentence “Typically, the biological method relies on macromolecular substances found in bacteria, fungi, and algae, such as enzymes, alcohol, flavonoids…” should be changed to plural

Response:

Alcohol is now changed into alcohols.

Page 8: Polymers and polysaccharides

- Delete the hyphen before D-glucose in the sentence “Starch (the capping agent) and -D-glucose (the reducing agent)….”

Response:

Hyphen before D- glucose is now deleted.

Reviewer 3 Report

It is a review based on 118 recent bibliographic titles.

The paper can be of real use to those who have research topics in the field.

I agree with the publication of the work, with the observation to standardize the writing of the bibliography.

Author Response

Reviewer 3:

It is a review based on 118 recent bibliographic titles.

The paper can be of real use to those who have research topics in the field.

I agree with the publication of the work, with the observation to standardize the writing of the bibliography.

Response:

Thank you very much to your positive review!

Reviewer 4 Report

Silver nanoparticles have aroused considerable interest due to their unique properties and proven applicability in various fields. Precisely, this topical topic is devoted to this review by Nguyen and Co. The review reports on various methods of obtaining nanoparticles, as well as a detailed consideration taking into account all the advantages and disadvantages. At times, the authors repeat themselves in the wording, however, this does not reduce the quality of the review. Therefore, I recommend this review for publication after making a minor revision.

1. Supplement the list of references with important sources in this field.(10.1016/j.arabjc.2014.12.014, 10.3389/fbioe.2019.00287, 10.1080/17458080.2016.1139196, 10.1080/21691401.2016.1241792)

2. Why are there no specifics in the examples of various types of biological activity? Add modern examples with antioxidant, antiviral, antidiabetic and other known types of activity for a comprehensive overview.

3. Two more important factors affecting the synthesis of nanoparticles and their stability are the pressure and concentration of AgNO3. Please add the appropriate items. (page 13)

Author Response

Reviewer 4:

Silver nanoparticles have aroused considerable interest due to their unique properties and proven applicability in various fields. Precisely, this topical topic is devoted to this review by Nguyen and Co. The review reports on various methods of obtaining nanoparticles, as well as a detailed consideration taking into account all the advantages and disadvantages. At times, the authors repeat themselves in the wording, however, this does not reduce the quality of the review. Therefore, I recommend this review for publication after making a minor revision.

  1. Supplement the list of references with important sources in this field.(10.1016/j.arabjc.2014.12.014, 10.3389/fbioe.2019.00287, 10.1080/17458080.2016.1139196, 10.1080/21691401.2016.1241792)

Response: References have been now added and highlighted in the revised manuscript.

  1. Why are there no specifics in the examples of various types of biological activity? Add modern examples with antioxidant, antiviral, antidiabetic and other known types of activity for a comprehensive overview.

Response:

The examples on biological activities have been now added and highlighted in the revised version.

  1. Two more important factors affecting the synthesis of nanoparticles and their stability are the pressure and concentration of AgNO3. Please add the appropriate items. (page 13)

Response:

More information on these two factors “Pressure and concentration of AgNO3” have been now added. The revised part is highlighted.